# Numerical Simulation of a Single and Double-Rotor Impact Crusher Using Discrete Element Method

Murray M. Bwalya [1],* and Ngonidzashe Chimwani [2],*

1   School of Chemical and Metallurgical Engineering, University of Witwatersrand, Wits, Johannesburg 2050, South Africa
2   Institute of the Development of Energy for African Sustainability (IDEAS), Research Centre of the University of South Africa (UNISA), Florida Campus, Private Bag X6, Johannesburg 1710, South Africa
*   Correspondence: Mulenga.Bwalya@wits.ac.za (M.M.B.); ngodzazw@gmail.com (N.C.)

**Abstract:** The Discrete element method (DEM) is an invaluable tool for studying comminution as it provides detailed information that can help with process analysis as well as trying out new equipment designs before the equipment is physically built. The DEM was used to analyse previous experimental work to gain some insight into the comminution process in an impact crusher with a single impeller. Further DEM simulations were done on a crusher with a second impeller installed. The energy spectra and threshold energy levels calculated from the drop-weight test were used as the basis of comparison. The simulations indicate that even at much lower speeds, the performance of a double impeller impact crusher is exceedingly superior. However, the energy associated with the double impeller impact crusher is much higher and energy intensification, rather than energy efficiency, is the main gain of the double impeller design. The double impeller also offers more operational flexibility, such as spacing between the impellers, which can be tailored to the particle size range being handled.

**Keywords:** discrete elemental method; impact crusher; single-rotor; double-rotor; counter-rotating; energy spectra; breakage model

## 1. Introduction

Comminution is a size reduction step that precedes physical separation, as valuable mineral grains must be freed by breakage from the gangue host matrix in order for them to respond to the separation process. This, however, comes at great energy expense, since particles have to be ground to sizes below 100 µm to achieve meaningful liberation. Unfortunately, the larger amount of energy applied to the comminution process does not contribute to particle breakage, but is rather wasted on processes such as driving the mechanical parts of the equipment, heat, and sound energy [1]. Comminution comprises crushing and grinding, but the former has been proven to be 10 to 20 times more energy-efficient than the later [2]. Therefore, overall optimisation of comminution can probably be achieved by increasing the contribution of crushers to the overall comminution process. Generally, grinding has received more attention than crushing over the years, thus possibly missing out on possible gains which can be achieved by improving crushing [3].

Typically, experimental approaches were mostly used in the design, optimisation, and customisation of crushers [4], however, this approach could not avail internal information regarding the flow of particles and how they break. Thus, researchers have resorted to computer modelling using advanced computational tools such as the discrete element method (DEM), which allows the simulation of particle flows in various equipment to be performed with a detailed prediction of the breakage process and a lot of other customizable information [3,5].

DEM is an advanced numerical computational tool that is increasingly being applied to solve particulate problems since Cundall and Strack [6] demonstrated its viability in soil

mechanics about four decades ago. A decade later, Mishra and Rajamani [7] adapted the DEM scheme to milling in mineral processing and since then its usage in comminution [8,9] has continued to expand. The ins and outs of this computational tool are well documented in many papers [4,8,9], which demonstrate its flexibility in addressing a wide range of particle related problems.

Despite the extensive usage of DEM in modelling particulate systems, particularly milling, a lot of work remains outstanding in crusher simulation as there is a wider variation in crusher equipment, each with its own consequential effect on particle breakage. A sample of the scanty research work that has been conducted in this field includes simulation of the horizontal impact crusher by both Djordjevic et al. [10] and Schubert et al. [11], albeit with a few particles and simplified geometries. Cunha et al. [12] investigated the distribution of collision energies and the residence time distribution in the same vertical shaft impact crusher. Later, Grunditz et al. [13] investigated the effect of feed size on the number of impacts and distribution of energy in the same crusher. Cleary [4] introduced a replacement strategy for breakage in which once the breakage criteria of a particle are met, the particle is replaced by progeny particles of sizes such that an original volume of the parent particle is covered satisfactorily. The replacement strategy was then later used to generate DEM models for the jaw crusher and vertical shaft impactor [14] and to predict the throughput and the product size distribution from the DEM model in the cone crusher [15]. Using the same replacement strategy, Sinnott and Cleary [5] successfully predicted power draw, throughput rate, PSD, and the equipment wear rate.

Cleary and Sinnott [3], and Sinnott and Cleary [5] rigorously studied the behaviour of both the compression-type crushers and impact-type crushers, respectively. In both studies, they used the linear-dashpot contact model, to calculate contact energy dissipations and resolve forces between the interacting elements. An in-depth discussion of the application of the linear-dashpot contact model in DEM simulation has been given by Navaro et al. [16] who concluded that the model generally gives accurate results.

With the valuable insights drawn from the foundational work done on systematic particle-based modelling for a wide range of crushers [3,5], this study focuses on the design factors that can improve the performance of an impact crusher. Using the DEM outputs such as spectral energy of inter-collisions and visual particle activity, the impact of a number of configurations are evaluated. The breakage rate prediction scheme, which will be a subject of another paper, is not developed here, as more experimental validation data will be required after the features highlighted by the simulator are tested by physical experimentation.

## 2. Methodology

### 2.1. DEM Simulation

Following promising results obtained in a single impeller crusher that involved repeated crushing of the product [17], the crusher design was improved by introducing a second impeller so that particles encounter two comminution stages before descending to the bottom of the device. Before implementing the new design, a prior investigation of the performance to be expected with this new design was made using the DEM tool.

The crusher geometry and the set-up of the simulation were done using the PFC3D (version 2.1, Itasca Group, Inc., Minneapolis, MN, USA) [18] which has an internal programming language called Fish. The model parameters are presented in Table 1. The impact crusher design and the rendering of its simulated version are shown in Figure 1, with dimensions specified in Table 2. The results of the physical experiments that were conducted with a single impeller crusher have been reported in a recent paper [17].

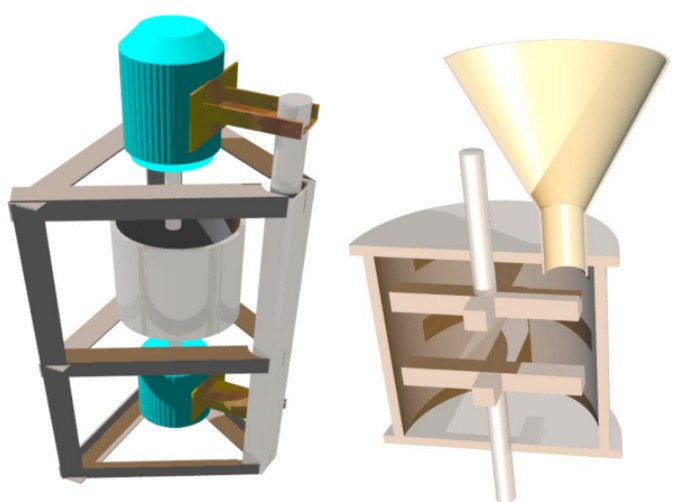

**Figure 1.** Design of the impact crusher and the rendering of its simulated version.

**Table 1.** Model parameters.

| Parameter | Value | Units |
|---|---|---|
| Kn | $4 \times 10^6$ | $N \cdot m^{-1}$ |
| Kt | $3 \times 10^6$ | $N \cdot m^{-1}$ |
| Wall friction coefficient | 0.4 | - |
| Particle friction coefficient | 0.4 | - |
| e | 0.2 | - |
| Cn | 0.6 | $N \cdot m^{-1} \cdot s$ |

**Table 2.** Specifications of the crusher.

| Specifications | Parameters | Values |
|---|---|---|
| Dimensions | Internal height | 0.20 m |
| | Internal diameter | 0.216 m |
| | Volume | 0.00733 $m^3$ |
| Impeller configuration | Impeller dimensions | 0.184 m × 0.179 m |
| | Impeller bar dimensions | 0.02 m × 0.02 m |

The single-impeller impact crusher simulations were based on three size classes: $-19.0 + 16.7$ mm, $-13.2 + 11.20$ mm, and $-7.92 + 5.60$ mm. For each size class, three impeller speeds were used: 510 rpm, 1040 rpm, and 2080 rpm. For the double impeller, on the other hand, only 1040 rpm speed was simulated for size classes $-19.0 + 16.7$ mm and $-13.2 + 11.20$ mm, for the purpose of comparing with single impact impeller. The power drawn and the energy spectra resulting from the particle collision events were recorded.

*2.2. Simulation Parameters*

The entity contact laws were based on the linear spring-dashpot model, with the normal stiffness and shear (or tangential) stiffness parameters being based on previous work done by Bwalya [19]. Table 1 shows the parameters used. The simulations time step was based on Equation (1) using the smallest particle in the system [20]:

$$\delta t = 2\pi \sqrt{\frac{m}{K}} \tag{1}$$

where *m* is the mass of the smallest and lightest particle (kg) and *K* is the particle normal stiffness. In our experience, it was found that a smaller time step is required when the

difference between the largest and smallest particle is big; thus, to suit our purpose, the $2\pi$ factor was replaced with 0.2.

### 2.3. Breakage Criterion

The breakage criterion was based on the drop-weight probability breakage model developed by Bwalya and Chimwani [21], which states that the probability of a particle breaking depends on the number of attempts, $n$, made to break the particle and the energy of each of those attempts, $E$. The relationship is summarised by the following equation:

$$P_b = 1 - e^{cn^d \left( \frac{E - E_{x0}}{E_{x0}} \right)} \tag{2}$$

where $P_b$ is the breakage probability for a particular particle, $n$ is the number of impact events a particle is exposed to, while $c$ and $d$ are parameters related to the deterioration of the particle with each successive impact depending on its material characteristics. $E_{xo}$ is the minimum impact energy (threshold energy) that must be exceeded before there can be any possibility of particle damage which would subsequently lead to breakage. This energy is calculated using Equation (3) and a methodology to obtain the $E_{x0}$ value from experimental data is given [21]:

$$E_{x0} = am^b \tag{3}$$

where $m$ is the mass of the particle in grams while $a$ and $b$ are material-specific parameters that define how particle strength is related to its size.

The breakage probability in Equation (2) is a general model that applies to incremental damage as well as single impact breakage. Thus, for any material, a suite of drop weight tests must be conducted, as discussed by   parameters.

## 3. Results and Discussion

The evaluation of the impact crusher's performance was based on the recorded energy dissipation in each of the iteration events in the simulated period. As pointed by Cleary and Sinnott [3], the energy spectra can be used to characterise a comminution environment and show how design changes or operational mode can influence expected performance. The changes in the energy spectra were assessed under different modes of operating the impact crusher. The scenarios explored involve: effect of rotor speed, particle size effect and rotor configuration. To compare simulations of different time durations, all the events were normalised to a time duration of 1 s.

### 3.1. Rotor Speed Effect

The rotor speed of the impacting impeller can be varied and, as reported in recent work by Chimwani and Bwalya [17] covering a range of experiments, speed affect milling rate significantly. With the DEM, the change in operation conditions translates into variation in the energy spectra record and is an indirect indicator of actual breakage that can be expected for the simulated condition. This will, however, also depend on the material breakage characteristics as defined by Equation (3). It must also be realised that the events each particle is subjected to in a comminution environment are random, and that will be the subject of another paper that will implement the breakage scheme. The present work will just make note of recorded energy events that meet the minimum required particle breakage energy, often referred to as threshold energy. If the energy of the impact is smaller than the threshold energy, the particle is expected to never break no matter how many times such impacts are repeated. Above the threshold energy with just a small margin, the particle will only break after many repeated attempts, whereas if the margin is big, particle breakage can be achieved with fewer breakage attempts. Equation (3) is used to calculate the threshold energy for any material and the parameters in the equation are determined experimentally using the drop-weight tester [21]. In this paper, the parameters that were determined for two materials, silica and Dolomite are given in Table 3. The table

also includes the threshold energy levels for the particles that are simulated. The particle masses in the table have been calculated based on the lower screen size as follows:

$$\pi \frac{d_L{}^3}{6} \rho \tag{4}$$

where $d_L$ is the lower screen size of a screen fraction and $\rho$ is the density of the material.

**Table 3.** Calculated $E_{x0}$ values based on previous work [21].

| Screen size | Silica (a = 0.3 b = 0.99) Sg = 2.65 | | Dolomite (a = 0.33 b = 0.37) Sg = 2.85 | |
|---|---|---|---|---|
| | Mass (g) | $E_{x0}$ | Mass (g) | $E_{x0}$ |
| −19.0 + 16.0 mm | 5.68 | 1.68 | 6.11 | 0.65 |
| −13.2 + 11.2 mm | 1.95 | 0.58 | 2.10 | 0.43 |
| −9.5 + 8.0 mm | 0.71 | 0.21 | 0.76 | 0.30 |

Based on this table, all the events that are above the threshold energy were calculated depending on the material or particle size.

Figure 2 shows the energy spectra trend with doubling impeller speed. It is observed that although there are thousands of contact events, only a few of these involve energy levels above particle threshold energy that can possibly result in particle breakage. It is, however, surprising to note that for 510 rpm speed all the energy levels appear to be below the threshold, indicating that no breakage would occur at this slow speed while in actual experimental work, the breakage was over 10%. There are several reasons to account for this; firstly, the simulator handles particles as spheres which cannot accurately represent a screen fraction of particles of a different shapes and distribution. Thus, $E_{x0}$ should be viewed as a statistical indicator of minimum energy required to cause particle failure. It is, however, clear that increasing impeller speed shifts the graphs to the right, hence the reason why the breakage rate increases with impeller speed. It is also observed that although larger particles in a simulation involve a smaller number of particles, the reduction in the number of recorded events does not correspond accordingly. This is due to the relatively long time that the particles spend in the collision space. The full explanation of this phenomenon is discussed by Chimwani and Bwalya [17].

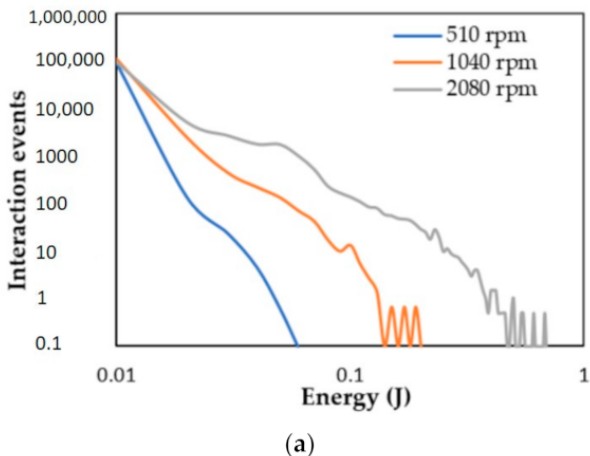

(a)

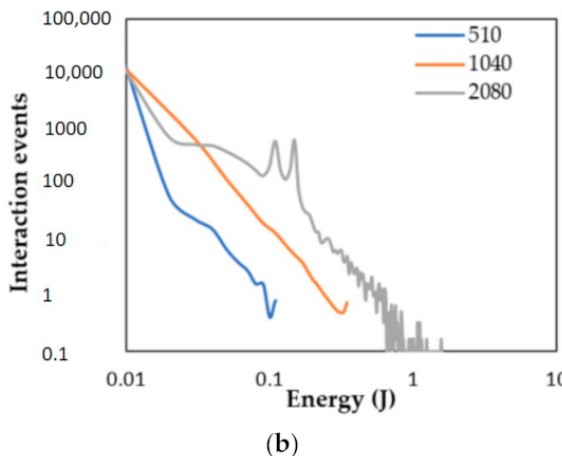

(b)

**Figure 2.** Energy spectra for the effect of rotor speed on single rotor design, particle size (**a**) −13.2 + 11.2 and (**b**) −19.0 + 16.7.

It must also be pointed out that, in the Chimwani and Bwalya [17] paper referred above, a drop-weight tester was used for breakage characterisation and cannot accurately match the impact crusher breakage conditions, hence the need to carry further investigations in future.

The energy spectra as presented in Figure 3 show a significant increase in the number of collisions for all rotor speeds. It can be seen that all rotor speeds produced higher energy impacts with a significant percentage of the collisions having energy beyond the minimum required for breakage, i.e., the threshold energy. This upsurge is owing to increased surface area for collisions provided by the additional impeller. Within a confined space, particles with high energy seldom lose all their energy in a single collision but rather ricochet across any surface in their path. The higher the velocities of the particles, the higher the probability of ricocheting across multiple surfaces, and that consequently raise the number of impacts and collisions with energy above the threshold energy. Therefore, an increase in rotor speed increases particle breakage, since the number of higher-energy impacts (greater than the threshold energy) of a particle increases the probability of breakage according to Bwalya's [19] model.

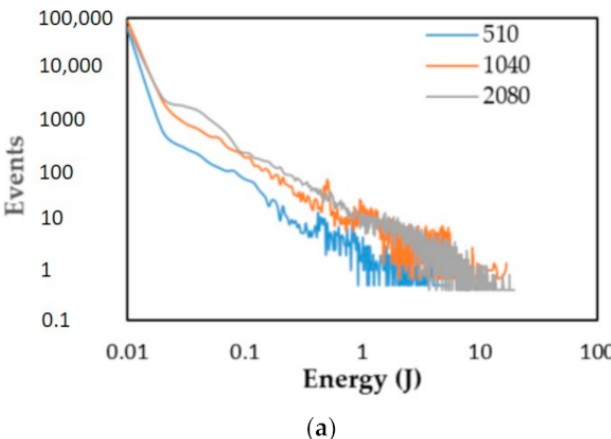 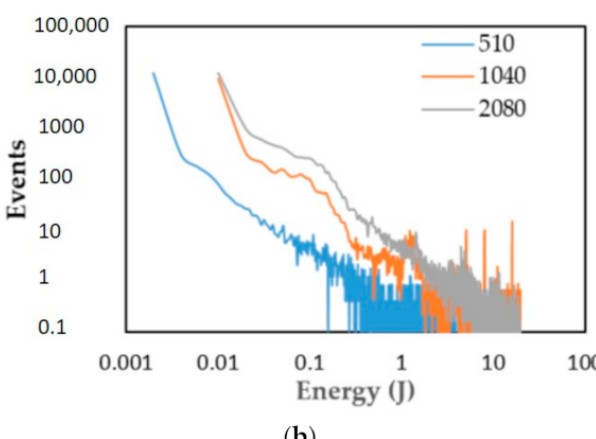

(**a**)　　　　　　　　　　　　　　　　　　　　(**b**)

**Figure 3.** Energy spectra for the effect of rotor speed on double rotor design, for particle size (**a**) −19.0 + 16.7 and (**b**) −13.2 + 11.2.

### 3.2. Particle Size Effect

The rotor speed of 1040 rpm was used to investigate the effect of particle size on the number of impacts achieved and the energy associated with those impacts. The energy spectra of the collisions of the particle size ranges for single and double impellers are presented in Figure 4. Three sizes, −19 + 16.7 mm, −13.2 + 11.2 mm, and −9.5 + 8 mm, are compared for both single and double rotors. While the total number of events is roughly in the ratio 25:7:1 for big: medium: smallest sizes, respectively, as expected due to particle numbers involved, the distribution of the energy levels is however surprising. For single-rotor speed, the −9.5 mm + 8 mm size range displays more higher energy events than the −13.2 + 11.2 mm, while for the double rotor, the −13.2 + 11.2 mm seem to have the highest energy events. Though the reason for this is no clear, it is possible that the relative spacing of the rotors on the shaft might affect particle collisions differently and future simulations and experiments should investigate optimal rotor spacing for specific particle sizes.

### 3.3. Effect of Number of Rotors

Instead of the one rotor that was initially used in the previous experiments [21], a second counter-rotating rotor was introduced in the simulation to determine if this design would enhance overall performance. Figure 5 shows how the single rotor energy spectra compare with the double rotor's spectra, for particle size −13.2 + 11.2 mm at 1040 rpm and 2080 rpm. For both speeds, the first simulation comprised one rotor at the set speed followed by two counter-rotating rotors at that speed. The observed improvement is phenomenal, especially in terms of the number of events with energy above the threshold level.

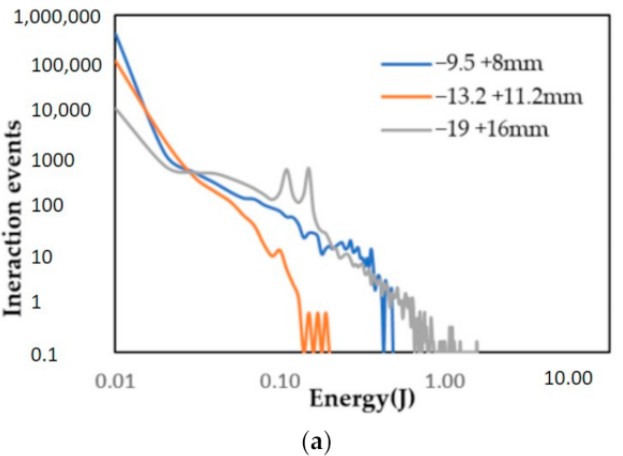
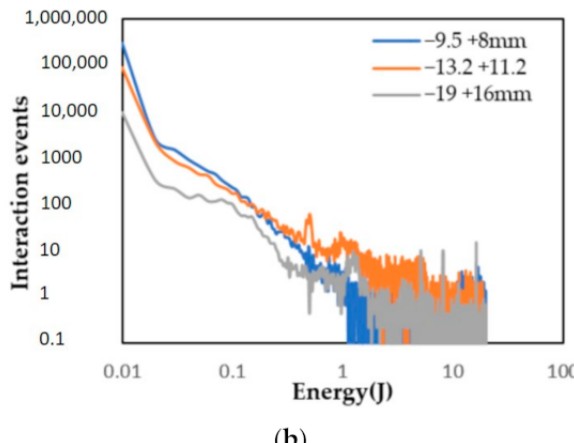

**Figure 4.** Energy spectra for the effect particle size on (**a**) single rotor and (**b**) double rotor design for 1040 rpm rotor speed.

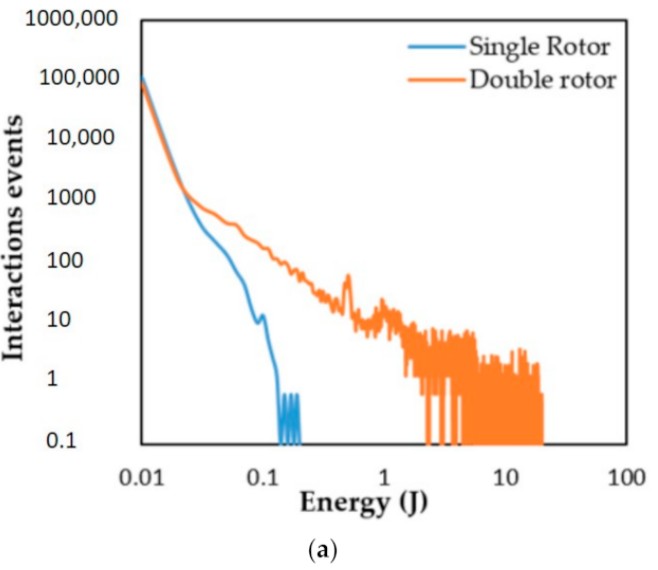
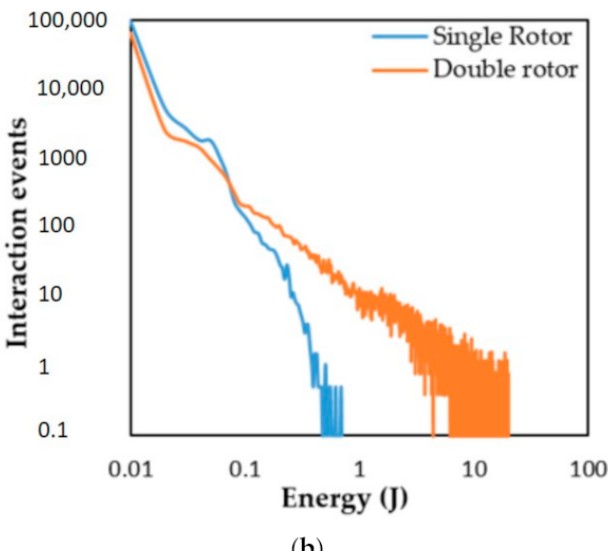

**Figure 5.** Energy spectra Comparison for single and double rotor for 13.2 + 11.2 mm particles; (**a**) at 1040 rpm and (**b**) at 2080 rpm.

The effect of using an additional rotor is also investigated for the −19 to 16 mm particles, but this time, the effect of co-rotation of the impellers is also tested. It is seen in Figure 6 that a second co-rotated rotor is significantly more effective in generating high energy impact events than a single rotor, but counter-rotation is the most effective.

The observed enhancement with counter-rotation over co-rotation is a study that will require work as this was based on one particular position of the second rotor relative to the first. At this point, it is assumed that particles projected by the first impeller will have higher energy of collision when they encounter the second impeller rotating in the opposite direction. However, some of the particles are projected directly onto the crusher wall and, thus,, their direction will be already reversed by the time they encounter the second impeller. This will be an issue that will be tackled in future when equipment optimisation and wear problems are addressed.

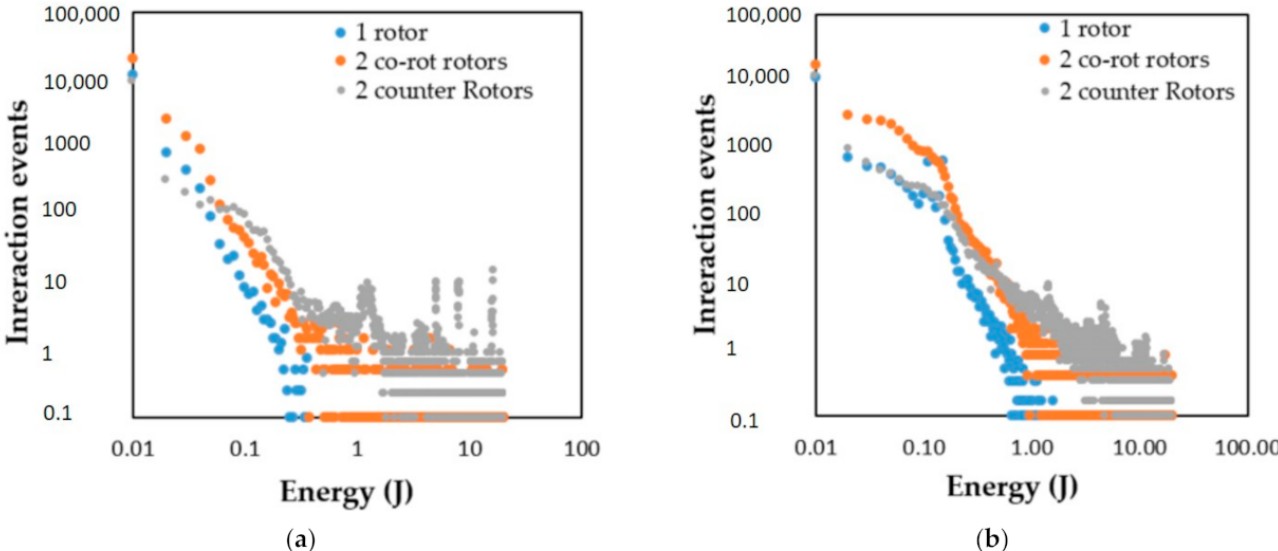

(**a**)
(**b**)

**Figure 6.** Comparison of the single rotor with double co-rotated and double counter-rotated rotors at (**a**) 1040 rpm and (**b**) 2080 rpm.

### 3.4. Combined Effect

With the foregoing observations, it has been established that the double rotor impact crusher has greater performance than the single rotor impact crusher under all conditions, albeit also consuming the most power. As seen in Figure 7, for the three configurations, single rotor, co-rotation rotor, and counter-rotation rotor, the increased performance with double impeller corresponds to higher power draw, suggesting that energy intensification rather than energy efficiency is the main benefit of introducing a second impeller.

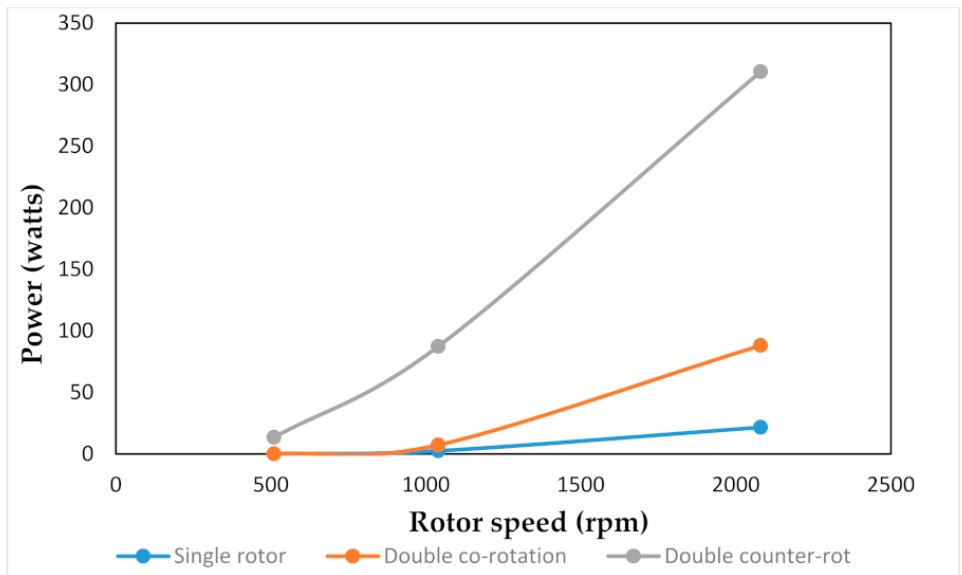

**Figure 7.** Power consumption under different modes of operation and speeds.

Comparing the effects of increasing rotor speed and the addition of the second rotor at the lowest speed has revealed that the lowest speed of the two-rotor design outperformed the highest speed of single-rotor design, and the comparison is depicted in Figure 8. Although the high-speed single rotor produced more impacts than a double-impact impeller, most of the impacts were not useful and only a few had high energy as well as energy greater than the threshold energy.

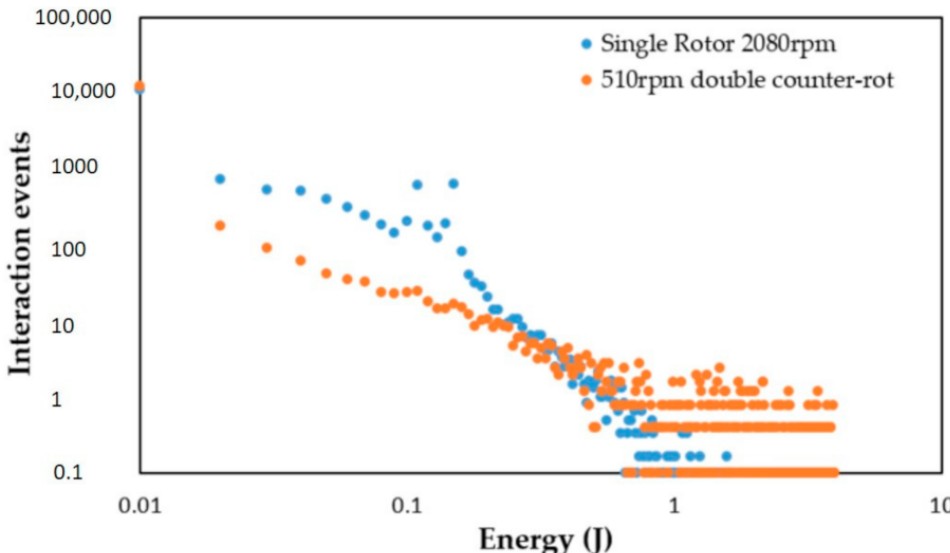

**Figure 8.** Energy spectra comparing single-rotor at 2080 rpm and double-rotor at 510 rpm, particle size 19.0 + 16.7 mm.

Table 4 presents the number of events and useful events according to the threshold energy of 0.747 J, for Dolomite particle size −19.0 + 16.7 mm. It can be seen that the difference in the number of impacts with energy above the threshold energy between the double and single rotors is very significant.

**Table 4.** Number of events and useful events at different rotor speeds.

| | | Number of Useful Impacts | |
| Rotor | Speed | Total Impacts | Impacts with Energy > Threshold Energy |
|---|---|---|---|
| 1 | 2080 | 225,687 | 23 |
| 2 | 510 | 147,974 | 3536 |

It can, thus, be seen that if increasing particle breakage in the crusher is achieved by increasing the rotor speed, not only does it increase the operational costs but also leads to inefficient breakage of particles. High operational costs in the sense that the higher rotor speed has maintenance implications as the impeller is automatically subjected to higher impact forces. However, the energy-saving factor is not clear cut, as observed in Figure 7, and the main advantage of the double rotor is the higher energy intensification that is easily achieved. This was explored further by analysing how the crushing environments are contrasted between single and double rotor configurations.

*3.5. Visual Comparison of the Single and Double Rotor Configurations*

Snapshots of particle positions were taken at earlier and later stages of the simulation to compare the two environments and are shown in Figure 9.

It is clearly seen in Figure 9 that the double rotor tends to cluster particles in the upper region as the second impeller acts as an additional obstacle that ricochets particles that are trying to gravitate downwards. This results in enhanced inter-particle collision events as particles interact in a smaller space. Additionally, the extra obstacle provided by the second rotor in the crusher enhances particle collision in the crusher by keeping particles in suspension for a longer time before they settle at the bottom of the crusher.

The distribution of average speeds for the entire simulation, however, paints a different picture, based on the sampled region shown in Figure 10 and the average velocities obtained over the simulation period as shown in Figure 11.

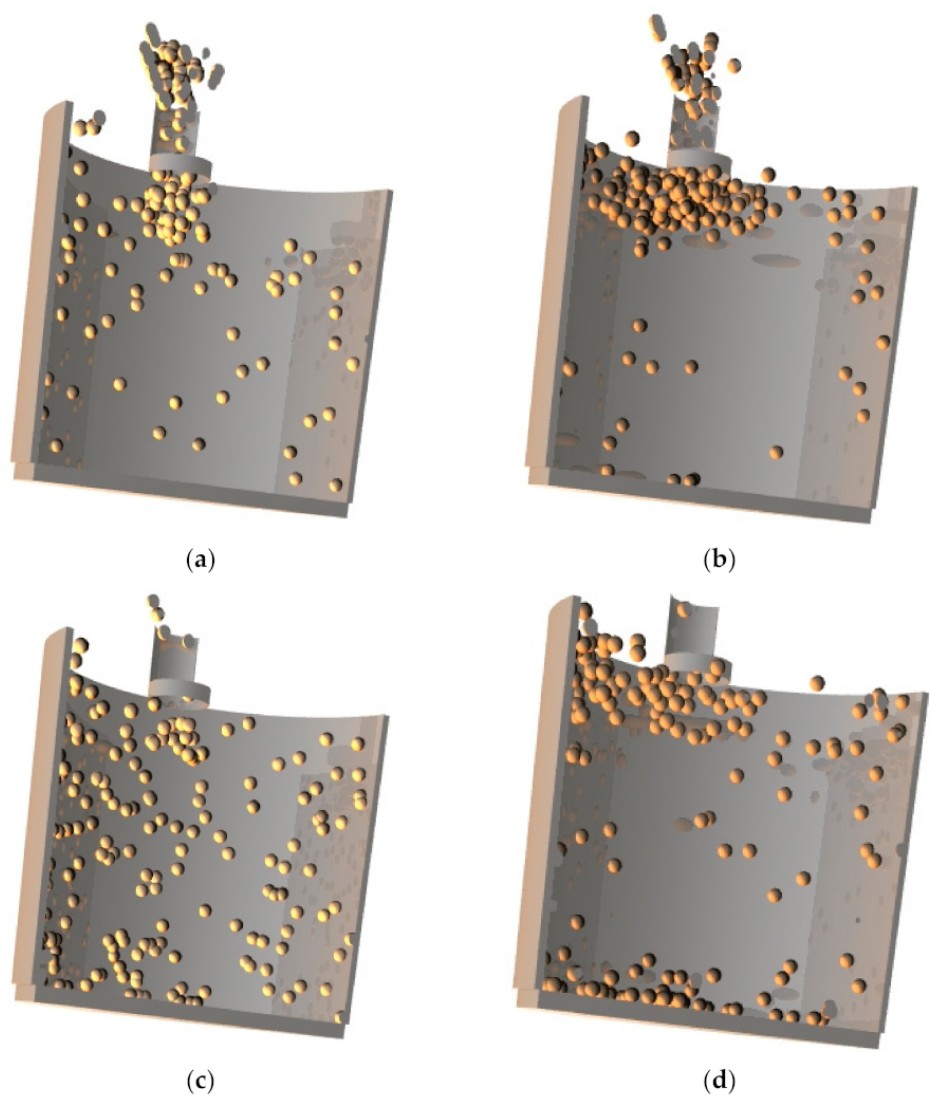

**Figure 9.** Snapshots of particle positions of the −9.5 + 8.0 mm for single and double rotor environments both operating at 1040 rpm impeller speed. (**a**) Frame 30 single rotor. (**b**) Frame 30 double rotor. (**c**) Frame 60 single rotor. (**d**) Frame 60 Double rotor.

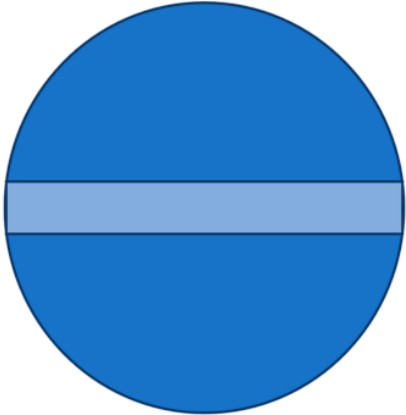

**Figure 10.** Highlighted area showing cross-section of the crusher sampled to obtain average speeds.

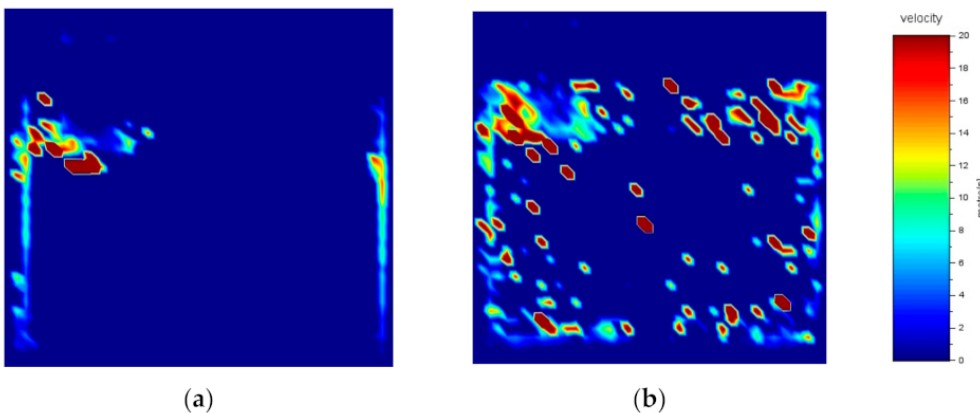

**Figure 11.** The distribution of particle speeds for the (**a**) single and (**b**) double rotor crushing environments, at 1080 rpm impeller speed.

It is seen that for the single rotor, highest speeds are mostly on the wall surfaces, indicating that most grinding will occur as a result of the interaction of particles with the wall and, thus, proposing a future design consideration for performance enhancement. The highest speeds are, however, observed around the point where the particles enter the crusher due to collisions between particles entering the crusher and those ricocheted by the impeller. A similar pattern was observed for the double rotor, but the action map gets extended throughout the crusher, suggesting that there are more inter-particle collisions with this later arrangement. If this is the case, which is yet to be verified experimentally, then the introduction of the second impeller also mitigates wear to some extent as the higher percentage of comminution is due to inter-particle collisions.

The comparison above is based on one particular spacing of the two rotors for the double rotor configuration and relative positioning of the rotors for the different particle sizes is a feature that will be explored in future.

## 4. Conclusions

DEM has been used to model the impact crusher performance for single and double rotors. The scheme of applying energy spectra and particle threshold energies has provided insightful information indicating performances of the various configurations under review. As observed from actual experimental work previously carried out, the simulator shows that increasing impeller speed increases the number of collision events that are above particle threshold energy for all the different particle sizes and, thus, increases breakage. However, for reasons not fully understood, the simulator predicts no breakage for some lower impeller speeds, when breakage actually occurred in the experimental work, and this will, thus, be one of the issues that will be addressed in future research work. It has also been shown, as expected, that introducing a second impeller enhances the breakage rate. However, what is surprising, is the order of improvement; apparently, a series of counter-rotating impellers at only 510 rpm will perform better than a single impeller at 2080 rpm. This does not necessarily mean that double impeller configuration is more efficient, as correspondingly higher energy consumption is also associated with this kind of configuration. Thus, the real benefit is the energy intensification that becomes possible with the introduction of an extra impeller. In this paper, threshold energy was used as a quick way to compare potential comminution capacity; however, a future paper with a detailed breakage scheme is planned to address this issue thoroughly. This will, however, come after more verification data are available after the newly designed equipment is commissioned.

**Author Contributions:** Conceptualization, N.C. and M.M.B.; methodology, N.C. and M.M.B.; software, M.M.B.; validation, N.C. and M.M.B.; formal analysis, N.C. and M.M.B.; investigation, N.C. and M.M.B.; resources, N.C. and M.M.B.; data curation, N.C. and M.M.B.; writing—original draft preparation, N.C. and M.M.B.; writing—review and editing, N.C. and M.M.B.; visualization, N.C. and M.M.B.; supervision, N.C. and M.M.B.; project administration, N.C. and M.M.B. All authors have read and agreed to the published version of the manuscript.

**Funding:** This research was funded by the National Research Fund (NRF) South Africa, Development Grant for Y-rated Researchers (Grant No 120394).

**Data Availability Statement:** Data supporting reported results can be provided from authors upon request.

**Acknowledgments:** The authors wish to thank the NRF, the University of the Witwatersrand, and the University of South Africa for supporting this work.

**Conflicts of Interest:** The authors declare no conflict of interest. The funders had no role in the design of the study; in the collection, analyses, or interpretation of data; in the writing of the manuscript; or in the decision to publish the results.

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
