# Peer review of "Numerical Simulation of a Single and Double-Rotor Impact Crusher Using Discrete Element Method"

_minerals, doi:10.3390/min12020143_

Round 1
Reviewer 1 Report
#General
The authors present a DEM simulation framework for comparing the energy spectra of two impact crusher designs under different operating conditions. The authors base their simulation on a laboratory-scale crusher presented in a previous publication. It is my impression that given the already existing and published experimental work, this paper requires further work. More specifically it requires to include breakage of the particles, so results can be compared against experimental data, and thus validate the energy spectra obtained via simulation.
The authors claim that previous work has shown that if the energy is below Ex0 then the particle will never break. In their simulations, especially with 1 impeller, very few impacts are above that energy so one should expect very little breakage. Does this correlate to what was observed in the experimental work? I reckon this is why breakage should be included in this work in order to sustain the energy spectra obtained.for example, in the range -19+16mm for silica Ex0 is 2.16 and in Figure 2a there is no collision with an energy above 1, thus no breakage should happen whatever the speed of the impeller. Almost no explanation is given for these results.
The paper reads well, but requires some editing. See below.
#Specific
The abstract mentions that DEM could provide insights into determining if adding a secon impeller would translate in “significant” improvements. It is not clear and explicit if the results are indeed “significant”. Authors mentioned an increased number of impacts of higher energies, but they also show an increased power draw of the equipment.
Line 27: put commas around “however”
Line 73-74: this sentence is out of place. There’s no link to the previous sentence and the theme of the paragraph
Line 91: impact “mill” or “crusher”
Line 97: there is a minus sign missing for the range -19+16mm
Line 101: ball or particle?
Line 107: should read only “ (1)”
Table 1: Ball or particle?
Line 165: should be Equation 3
Line 225-227: could the authors provide more explanation on why the effect described?
Line 249: -19+16mm
Author Response
Minerals
Milling studies in an impact crusher I: kinetics modelling based on Population balance modelling: minerals-1484084
Dear Reviewer
RE: LIST OF CHANGES ON THE POINTS RAISED BY REVIEWER
Thank you for your useful technical comments and suggestions on our manuscript. We have modified the manuscript accordingly.
Detailed corrections are listed below point by point. Points raised by the reviewers are in italics and changes made or our response to those points in normal text.
Reviewer’s comments:
The authors present a DEM simulation framework for comparing the energy spectra of two impact crusher designs under different operating conditions. The authors base their simulation on a laboratory-scale crusher presented in a previous publication. It is my impression that given the already existing and published experimental work, this paper requires further work. More specifically it requires to include breakage of the particles, so results can be compared against experimental data, and thus validate the energy spectra obtained via simulation.
The authors claim that previous work has shown that if the energy is below Ex0 then the particle will never break. In their simulations, especially with 1 impeller, very few impacts are above that energy so one should expect very little breakage. Does this correlate to what was observed in the experimental work? I reckon this is why breakage should be included in this work in order to sustain the energy spectra obtained. For example, in the range -19+16mm for silica Ex0 is 2.16 and in Figure 2a there is no collision with an energy above 1, thus no breakage should happen whatever the speed of the impeller. Almost no explanation is given for these results.
- Theoretically, particle should not be broken, but the DEM is based on spherical shape, while for irregular shapes, it is possible for particles to be smaller than the expected range. Ex0 must also be understood as statistical probability. This point is now discussed in more details in the paper.
The paper reads well, but requires some editing. See below.)
- Thank you.
The abstract mentions that DEM could provide insights into determining if adding a second impeller would translate in “significant” improvements. It is not clear and explicit if the results are indeed “significant”. Authors mentioned an increased number of impacts of higher energies, but they also show an increased power draw of the equipment
- The DEM is indicating that you get energy intensification by introducing a second impeller but this may not necessarily translate into great efficiency and thus power measurement will be paramount for the equipment that will be commissioned to verify these predictions. This information will become clearer with further research.
Line 27: put commas around “however”
- Sentence rearranged as “This however comes at great energy expense, particularly….”
Line 73-74: This sentence is out of place. There’s no link to the previous sentence and the theme of the paragraph
- Sentence removed
Line 91: impact “mill” or “crusher”
- Mill changed to crusher, changes also effected for Figure 1 title and Table 2 title.
Line 97: there is a minus sign missing for the range -19+16mm –
- Correction done
Line 101: ball or particle?
- Correction done
Line 107: should read only “ (1)”
- Word ‘equation’ removed
Table 1: Ball or particle?
- Changed to particle
Line 165: should be Equation 3
- Corrected
Line 225-227: could the authors provide more explanation on why the effect described?
- Authors are suggesting rotor spacing as the possible reason and will study this more thoroughly in future.
Line 249: -19+16mm
- Corrected
The authors would like to thank the reviewer once more for the constructive comments that help to improve the quality of our work.
Kind regards,
Reviewer 2 Report
Mayor comments
- The novelty of the research must be highlighted
- In Comminution, an important parameter is the size of the product. An estimation of the particle size distribution of the product must be added with the proper justification.
- Some experimental validation is needed. None of the results are compared with experimental data, and a calibration process is not shown. Parameter calibration is required in DEM
- Snapshots or images of the DEM simulation with particles are missing. Only a 3D CAD model of the geometry is presented. Please add one or more snapshots of your DEM simulation. This might help the reader to understand your work.
- It is not clear how the breakage is modeled. A breakage probability is defined, and material parameters are defined, but there are not any results with breakage probability. Is the breakage implemented in the DEM model? Is it used a particle-replacement method or bonded-particle method?
Minor comments
- Line 1: it should be Article instead of Review
- Line 46: It is Strack
- Lines 84 to 88: A cite is required.
- Lines 97 and 99: the size class should be -19+16.7 mm, the minus symbol is missing.
- Line 107: Equation 1 is badly numbered.
- Lines 121 and Line 122: n and E are not formatted as equations, as well many other symbols. Please format all the equations and symbols as equations
- Lines 152 to 166: Some of that paragraph, should be in methodology instead of results.
- Table 3: breakage parameters must be in Methodology, not in Results and Discussion
- Line 172: it is missing a “m = “ or something similar in the equation
- Figure 7: There are necessary more comments about this figure.

Author Response
Minerals
Milling studies in an impact crusher I: kinetics modelling based on Population balance modelling: minerals-1484084
Dear Reviewer
RE: LIST OF CHANGES ON THE POINTS RAISED BY REVIEWER
Thank you for your useful technical comments and suggestions on our manuscript. We have modified the manuscript accordingly.
Detailed corrections are listed below point by point. Points raised by the reviewers are in italics and changes made or our response to those points in normal text.
Reviewer’s comments:
The novelty of the research must be highlighted The corrections were done; however, the authors are of the opinion that efficiency is the eventual goal which is achieved through many steps some of which when achieved prompt the use of the word.
- Our main highlight is the use of the DEM to predict expected performance of new equipment that has been designed and is about to be commissioned. We look forward to reporting the actual results when experimental work is done with the new design.
In Comminution, an important parameter is the size of the product. An estimation of the particle size distribution of the product must be added with the proper justification
- For this simulation, we were not breaking particles but the simulation output was adequate to compare the different scenarios that were studied. From experiments reported in previous paper (Bwalya. M., 2005), we were able to infer expectation of the new design that is about to be implemented.
Some experimental validation is needed. None of the results are compared with experimental data, and a calibration process is not shown.
- Parameter calibration is required in DEM – we rely on previous simulations for calibrations, but more detailed calibration will be required for our future paper that will implement a more detailed breakage scheme.
Snapshots or images of the DEM simulation with particles are missing. Only a 3D CAD model of the geometry is presented. Please add one or more snapshots of your DEM simulation. This might help the reader to understand your work.
- Snapshots have been added and velocity profiles included.
It is not clear how the breakage is modeled. A breakage probability is defined, and material parameters are defined, but there are not any results with breakage probability. Is the breakage implemented in the DEM model? Is it used a particle-replacement method or bonded-particle method?
- A more elaborate scheme is still in the works, here the authors want to emphasise how DEM has helped us to evaluate a proposed design of using a second rotor to improve on the previous 1 rotor crusher.
Line 1: It should be Article instead of Review –
- Changed
Line 46: It is Strack
- Corrected
Lines 84 to 88: A cite is required.
- Cited
Lines 97 and 99: the size class should be -19+16.7 mm, the minus symbol is missing.
- Corrected
Line 107: Equation 1 is badly numbered.
- Corrected
Lines 121 and Line 122: n and E are not formatted as equations, as well many other symbols. Please format all the equations and symbols as equations –
- Italicized to match equations
Table 3: breakage parameters must be in Methodology, not in Results and Discussion
- Table 3 presents calculations based on previous work; Table title changed and previous work cited.
Line 172: it is missing a “m = “ or something similar in the equation π
- Included
Figure 7: There are necessary more comments about this figure.
- Rewritten sections to emphasise that energy intensification is the main benefit of using double rotors.
The authors would like to thank the reviewer once more for the constructive comments that help to improve the quality of our work.
Kind regards,
Round 2
Reviewer 1 Report
The authors have addressed the comments made and have added clarity on how to interpret some of the results, especially those that do not seem to be backed by experimental results.
Minor edits to the text and figures are suggested in the attached document.
I am, however, still skeptical of the value of the results shown. It seems to me that the fact that breakage is not included makes all the results more like a "sanity check" of the simulations than providing actual insight into what is happening in the crusher. For example, more speed shows higher energy impacts, adding a second impeller adds more impact events, etc. All of these results are to be expected just from common sense (hence the sanity check feeling). Adding breakage will affect these results, and yet, there is no mention of adding breakage in the conclusions as future work.

Author Response
Minerals
Numerical Simulation of a Single and Double-Rotor Impact Crusher Using Discrete Element Method: minerals-1484084
Dear Reviewer
RE: LIST OF CHANGES ON THE POINTS RAISED BY REVIEWER
Thank you for your useful technical comments and suggestions on our manuscript. We have modified the manuscript accordingly.
Detailed corrections are listed below point by point. Points raised by the reviewers are in italics and changes made or our response to those points in normal text.
Reviewer’s comments:
The authors have addressed the comments made and have added clarity on how to interpret some of the results, especially those that do not seem to be backed by experimental results.
- Thank you.
Minor edits to the text and figures are suggested in the attached document.
- Thank you, we accepted all comments, before adding new material to the paper
I am, however, still skeptical of the value of the results shown. It seems to me that the fact that breakage is not included makes all the results more like a "sanity check" of the simulations than providing actual insight into what is happening in the crusher. For example, more speed shows higher energy impacts, adding a second impeller adds more impact events, etc. All of these results are to be expected just from common sense (hence the sanity check feeling). Adding breakage will affect these results, and yet, there is no mention of adding breakage in the conclusions as future work
- Thank you for these valuable observations; in fact the abstract and conclusion have been rewritten to highlight the real achievement of this work. Energy intensification is the main gain from the additional of an Impeller. While expecting higher performance with additional impeller is expected, logically, it is the step of the gain in performance that is baffling and for this reason we are expediting the commissioning of the equipment to validate the DEM prediction and get a better understanding of what is going on. The full breakage prediction scheme is also planned after we get more validation experimental data and this comment is now included in our paper.
Maybe conduct Single-Impact Load Cell measurements
- It will be very interesting to conduct Single-Impact Load Cell measurements but the authors are considering that for their future work. The authors have already discussed with Professor Miller at Utah University, the possibility of using his damage inspection technique using tomography and have started looking at how slow compression compares with drop-weight breakage for different materials.
Does experimental data support this phenomenon? Would you expect more breakage?
- Generally the bigger the particle size the higher the selection function. However a major outstanding work is the exploration of rotor position for both single and double rotor configuration. We believe if for a single rotor, the rotor were to be lowered than the current position, the results would substantially change, so we thus hope to have a more detailed investigation using both our DEM simulator as well as the actual equipment when it is commissioned.
The colormap makes hard to interpret, goes from green to blue to red twice.
- The figure was redone and the problem was fixed.
Not sure [10] and [11] are cited in the text
- The authors have fixed the references.
The authors really appreciate the reviewer’s effort to improve the quality of our work the
Kind regards,
Reviewer 2 Report
The observations were answered in a correct form by the authors. The article is good enough, but not outstanding.
Author Response
Dear Reviewer
The observations were answered in a correct form by the authors. The article is good enough, but not outstanding.
- Thank you, we have further improved the article.
Round 3
Reviewer 1 Report
I thank the authors for addressing the reviewer's comments and to better describe the the conclusions one can extract from their work. The "so-what" question is better described in the abstract and conclusions